# Total energy expenditure is repeatable in adults but not associated with short-term changes in body composition

Low total energy expenditure (TEE, MJ/d) has been a hypothesized risk factor for weight gain, but repeatability of TEE, a critical variable in longitudinal studies of energy balance, is understudied. We examine repeated doubly labeled water (DLW) measurements of TEE in 348 adults and 47 children from the IAEA DLW Database (mean ± SD time interval: 1.9 ± 2.9 y) to assess repeatability of TEE, and to examine if TEE adjusted for age, sex, fat-free mass, and fat mass is associated with changes in weight or body composition. Here, we report that repeatability of TEE is high for adults, but not children. Bivariate Bayesian mixed models show no among or within-individual correlation between body composition (fat mass or percentage) and unadjusted TEE in adults. For adults aged 20–60 y (N = 267; time interval: 7.4 ± 12.2 weeks), increases in adjusted TEE are associated with weight gain but not with changes in body composition; results are similar for subjects with intervals >4 weeks (N = 53; 29.1 ± 12.8 weeks). This suggests low TEE is not a risk factor for, and high TEE is not protective against, weight or body fat gain over the time intervals tested.

O besity is a highly prevalent health condition associated with increased morbidity and mortality[1]. The social and environmental factors behind the global obesity pandemic remain largely unresolved despite decades of research[2]. Nonetheless, the proximate cause of weight gain is an imbalance between energy intake and expenditure, making reliable measurements of total energy expenditure (TEE) an essential tool in medical and nutritional research. TEE is a critical variable in assessing energy balance and weight change, as TEE reflects the sum of energy expenditure on basal metabolic processes, thermoregulation, digestion, physical activity, and all other physiological tasks.

Low TEE has long been hypothesized to be a risk factor for obesity[3,4]. However, research in this area has produced mixed results. Early work using doubly labeled water (DLW) to measure TEE in free-living subjects found that women with obesity exhibited a similar body weight- and body composition-adjusted TEE compared to age-matched normal weight subjects[5]. Nonetheless, studies have reported that infants and children with low TEE gained more body fat than individuals with a higher TEE[4,6], but conversely that high TEE predicted a high rate of body fat gain in preadolescent girls[7]. Two studies reported that adults with a low 24-h energy expenditure (measured in a whole-room indirect calorimeter) were more likely to gain body mass over the subsequent 2.0–6.7 years than individuals with a high energy expenditure[3,8]. In contrast, several longitudinal studies have shown that TEE is not predictive of subsequent changes in body fat percentage in infants and children[9–12] or in adult women[13,14]. Energy expenditure measured with heart rate monitoring was found to be inversely associated with changes in fat mass (FM), but not in body weight, in participants younger than 54 years, and was positively associated with weight gain in participants older than 54 years[15].

One methodological consideration that may contribute to the conflicting nature of these findings is limited sample size. Changes in weight and adiposity under normal conditions (i.e., absent a dietary or other lifestyle intervention) are often slow, at the limits of detectability over short timescales. For example, average weight gain for U.S. adults aged 40–69 years is <1 kg/y[16]. Large samples are therefore needed to detect factors related to weight gain under normal conditions.

A second methodological factor that may adversely affect assessments of TEE and weight change is the reliability of TEE measurements. If TEE measurements fluctuate over time, due to physiological or behavioral changes or to measurement error, then a measurement at any given time point might not be reflective of average TEE over the period observed for weight change. Similarly, if TEE measurements are highly variable, differences between two time-points might not reflect durable, lasting changes in TEE but rather transient variation or measurement error. A combination of noisy data with small sample size may produce spurious results.

Basal metabolic rates and 24-h expenditures measured in calorimetry chambers have been reported to be repeatable in humans[17] and other animals[18–21], but in animals their repeatability declines as the interval between measurements increases[19,22–24]. Further, the condition under which animals live influences the repeatability of metabolic rate, which is lower for animals living under field versus laboratory conditions[23]. Less is known about the repeatability of TEE measurements. TEE was found to be repeatable when determined in the same five participants with a 3-day break between measurements[25]. Wong and colleagues[26] demonstrated repeatability of TEE measurements for 20–50-year-old adults (N = 50) for durations up to 2.5 years (Bland-Altman pair-wise comparison showed a lower and upper limit of agreement between −148 and 137 kcal/d and a paired

t test showed no difference between repeated measures of TEE P ≥ 0.3). In that study, the time that elapsed between both TEE measurements ranged from 12 days to 2.5 years, and almost 68% of those measurements were repeated within 1 year[26]. To date it is unclear if repeatability of TEE is related to the duration between TEE measurements, or whether repeatability differs for children or for adults older than 50 years.

This study had the following aims: (1) Determine whether subject age affects repeatability of TEE, and (2) examine if TEE is associated with changes in weight or body composition. We used The International Atomic Energy Agency (IAEA) DLW database (https://doubly-labelled-water-database.iaea.org/home, database version 3.1.2) which pools DLW data across multiple studies[27]. The database contains 6,787 measures of TEE spanning individuals from 23 countries. All measures of TEE were estimated using a common calculation method[28] removing variation introduced by choice of equation. We included all individuals that were at least 1 year old and for which repeated TEE measurements were available (N = 696 TEE measurements of 348 adults and 114 TEE measurements of 47 children). We estimated repeatability 'R', also referred to as the intra-class correlation coefficient (ICC), using a mixed effects model framework, where R describes the relative partitioning of variance into within-group and between-group sources of variance[29–31].

We used two approaches to examine if TEE is associated with changes in weight or body composition of adults. Firstly, we used a multi-response model to decompose the covariance between TEE and FM on a between ($r_{ind}$) and within-individual ($r_e$) level, where $r_{ind}$ indicates whether individual mean values of traits correlate, and where $r_e$ indicates when the change in one trait between two time points is correlated with the change in another trait over the same period within an individual[32,33] (see Methods section). Thus, $r_e$ represents combined, reversible changes in traits that occur within an individual, and $r_{ind}$ reflects genetic and permanent environmental effects that are responsible for the association between the traits[32,33]. Secondly, we calculated a body size- and composition-adjusted TEE (see Methods section). We used linear models to test whether adjusted TEE is associated with changes in body weight and body fat percentage in all adults 20–60 y (N = 267 adults; time interval: 7.4 ± 12.2 weeks) and in a subset of individuals (N = 53 adults; 29.1 ± 12.8 weeks) for which the time between TEE measurements exceeded 4 weeks.

## Results

**Repeatability estimate R (ICC).** We calculated repeatability of adjusted TEE, which controls for body composition variables (FFM, FM), sex, and age (see Methods section). For adults and children together, adjusted TEE was repeatable (R = 0.54, SE = 0.035; CI = 0.472–0.608; $P_{LRT}$ < 0.0001, $P_{Permutation}$ < 0.001; Fig. 1a). However, repeatability differed markedly between adults and children. Adjusted TEE was repeatable for adults (R = 0.64, SE = 0.033; CI = 0.578–0.703; $P_{LRT}$ < 0.0001, $P_{Permutation}$ < 0.001; Fig. 1b) but not for children (R = 0.00, SE = 0.077; CI = 0.000–0.262; $P_{LRT}$ = 1.0, $P_{Permutation}$ = 1.0; Fig. 1c).

We calculated repeatabilities of body mass adjusted for the fixed effects of sex and age. Body mass was more repeatable than TEE for both adults and children. For adults and children together, body mass was highly repeatable (R = 0.96, SE = 0.004; CI = 0.952–0.967; $P_{LRT}$ < 0.0001, $P_{Permutation}$ < 0.001, Fig. 1d). Body mass was also repeatable for adults (R = 0.94, SE = 0.006; CI = 0.929–0.952; $P_{LRT}$ < 0.0001, $P_{Permutation}$ < 0.001; Fig. 1e) and children (R = 0.38, SE = 0.107; CI = 0.166–0.583; $P_{LRT}$ < 0.0001, $P_{Permutation}$ = 0.012; Fig. 1f) when analyzed separately.

Similarly, alternative analyses using body composition-adjusted TEE also found repeatability of TEE in adults but not in children

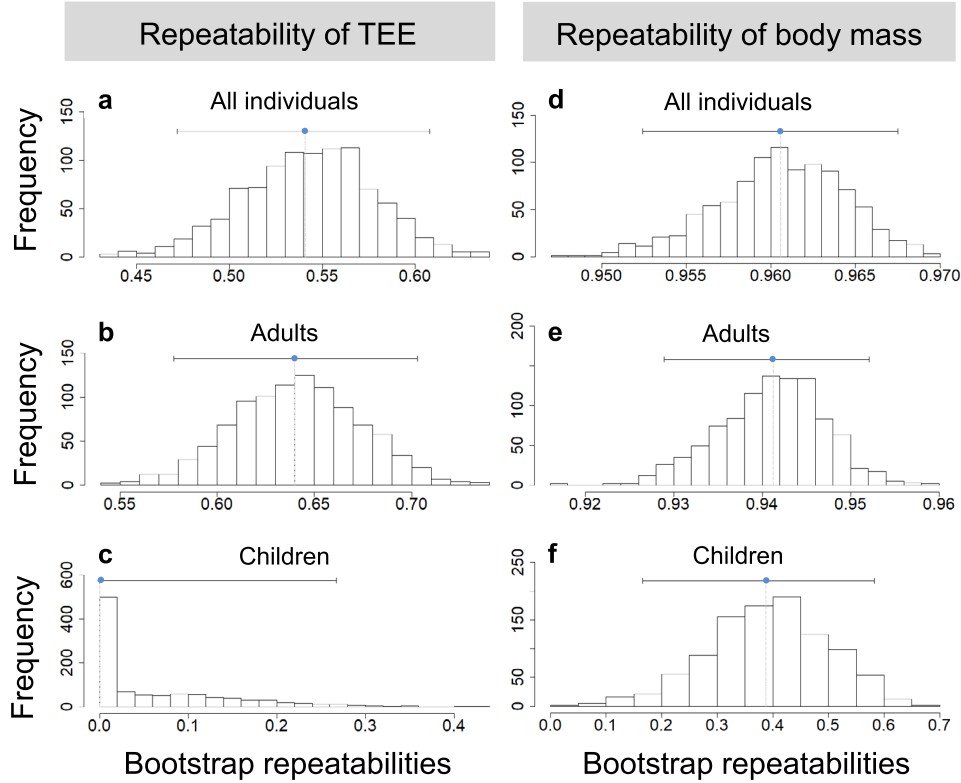

**Fig. 1 Repeatability estimates of total energy expenditure (TEE) and body mass.** Repeatability estimates of **a–c** TEE and **d–f** body mass at the individual level. Shown are distributions of the parametric bootstrap samples along with the point estimate of the repeatability estimate R (blue point) and the limits of the confidence interval (gray lines). **a**, **d** show all individuals together ($N = 395$ subjects), **b**, **e** show adults ($N = 348$ subjects) and **c**, **f** show children ($N = 47$ subjects) separately.

**Table 1 Phenotypic ($r_p$), among-individual ($r_{ind}$), and within-individual ($r_e$) correlations between unadjusted TEE and FM (Models $1 + 3$) and unadjusted TEE and body fat percentage (Models $2 + 4$).**

|   | Model | Traits | $r_p$ (95% CI) | $r_{ind}$ (95% CI) | $r_e$ (95% CI) |
|---|-------|--------|----------------|--------------------|----------------|
| A | Model 1 | TEE × FM | −0.07 (−0.16–0.04) | −0.09 (−0.21–0.05) | 0.04 (−0.10–0.16) |
|   | Model 2 | TEE × % FM | −0.04 (−0.14–0.07) | −0.01 (−0.15–0.12) | −0.05 (−0.19–0.06) |
| B | Model 3 | TEE × FM | 0.09 (−0.16–0.31) | 0.07 (−0.29–0.40) | 0.29 (−0.02–0.47) |
|   | Model 4 | TEE × % FM | 0.19 (−0.06–0.38) | −0.23 (−0.14–0.50) | 0.18 (−0.09–0.40) |

A shows result using the entire dataset ($N = 267$ subjects) and B shows the results using a subset ($N = 53$ subjects) for which the time between measurements exceeded 4 weeks. Correlations are presented with 95% credible intervals (CIs).
*TEE* total energy expenditure, *FM* fat mass, *% FM* body fat percentage.

(Supplementary Note 1 and Supplementary Fig. 1). Moreover, repeatability of TEE adjusted for FFM, FM, sex and age did not change with increasing time between both TEE measurements (Supplementary Fig. 1a–c).

**Is TEE associated with subsequent changes in weight or body composition?** In a first approach, we estimated the decomposition of covariances between unadjusted TEE and body fat (FM and body fat percentage) on a between ($r_{ind}$) and within-individual ($r_e$) level by multi-response mixed models (see Methods section). This analysis allowed us to differentiate between combined, reversible changes in traits that occur within an individual ($r_e$) and genetic and permanent environmental effects that are responsible for the association between the traits ($r_{ind}$). We used two datasets: the entire dataset of adults 20–60 y ($N = 267$ subjects; Models $1 + 2$) and a subset of individuals ($N = 53$ subjects) for which the time between measurements exceeded 4 weeks (Model $3 + 4$; Table 1). These models showed the same result: there is no within-individual ($r_e$), among-

individual ($r_{ind}$), or phenotypic ($r_p$) correlation between TEE and body fat (both as FM and body fat percentage; Table 1).

In a second approach, we calculated an adjusted TEE, accounting for the covariation of TEE with FFM, FM, age and sex (see Methods). Adjusted TEE was not correlated with short-term changes in body composition, and was correlated with change in body weight in only one analysis. Adjusted TEE1 (the first time point) was not associated with change in body weight (estimate ± SE: −0.001 ± 0.002, $t = -0.612$, df = 265, $P = 0.541$, adjusted $R^2 = -0.002$; Fig. 2a) or body fat percentage (estimate ± SE: 0.020 ± 0.017, $t = 1.206$, df = 265, $P = 0.229$, adjusted $R^2 = 0.001$; Fig. 2b). The difference in adjusted TEE between measurements was positively associated with changes in body weight, as subjects with greater adjusted TEE2 measures tended to weigh more (estimate ± SE: 0.009 ± 0.003, $t = 2.563$, df = 265, $P = 0.01$, adjusted $R^2 = 0.020$; Fig. 2c). Average adjusted TEE was not associated with changes in body weight (estimate ± SE: 0.001 ± 0.003, $t = 0.396$, df = 265, $P = 0.692$, adjusted $R^2 = -0.003$). Neither average adjusted TEE nor the difference in adjusted TEE between

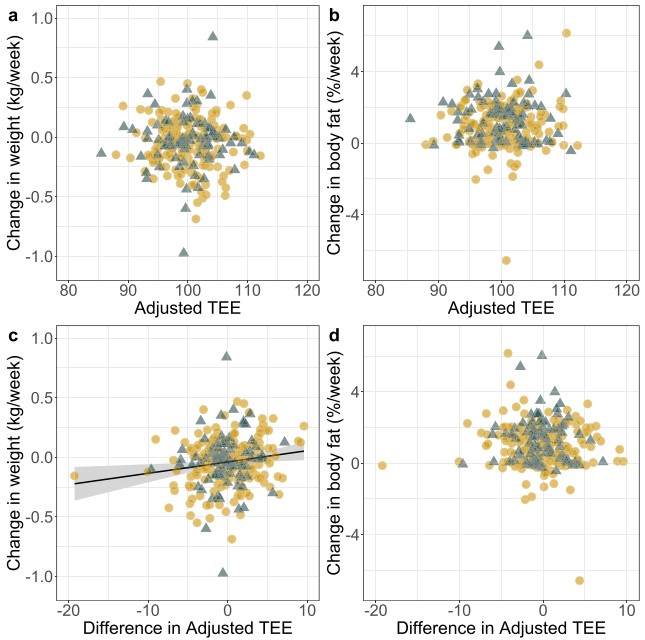

**Fig. 2 Relationship between adjusted total energy expenditure (TEE), the difference in adjusted TEE between measurements and changes in body weight and body fat percentage.** Relationship between adjusted TEE (MJ/d; adjusted for FFM, FM, age, and sex) at the first measurement and **a** changes in body weight and **b** changes in body fat percentage until the second TEE measurement, and the relationship between the difference in adjusted TEE between measurements (i.e., adjusted TEE2 – adjusted TEE1), and **c** changes in body weight (linear regression line is shown and shaded area indicates 95% confidence interval) and **d** changes in body fat percentage until the second TEE measurement ($N = 267$ subjects aged 20–60 years; yellow circles present females and gray triangles present males).

measurements were associated with body fat percentage (avg. adjusted TEE: estimate ± SE: $0.017 ± 0.019$, $t = 0.906$, df = 265, $P = 0.336$, adjusted $R^2 = -0.0006$; difference in adjusted TEE: estimate ± SE: $-0.023 ± 0.022$, $t = -1.042$, df = 265, $P = 0.298$, adjusted $R^2 = 0.0003$; Fig. 2d).

We obtained similar results in the analysis restricted to subjects with more than 4 weeks between TEE measurements. Adjusted TEE1 was not associated with change in body weight (estimate ± SE: $-0.004 ± 0.002$, $t = -1.704$, df = 51, $P = 0.094$, adjusted $R^2 = 0.035$; Supplementary Fig. 2a) or body fat percentage (estimate ± SE: $-0.0005 ± 0.006$, $t = -0.076$, df = 51, $P = 0.940$, adjusted $R^2 = -0.019$; Supplementary Fig. 2b) in these subjects. As in the full dataset, the difference in adjusted TEE between measurements was positively associated with changes in body weight (estimate ± SE: $0.010 ± 0.002$, $t = 4.278$, df = 51, $P < 0.0001$, adjusted $R^2 = 0.249$; Supplementary Fig. 2c). Average adjusted TEE was not associated with changes in body weight (estimate ± SE: $0.0003 ± 0.003$, $t = 0.100$, df = 51, $P = 0.921$, adjusted $R^2 = -0.019$). Neither average adjusted TEE nor the difference in adjusted TEE between measurements were associated with body fat percentage (avg. adjusted TEE: estimate ± SE: $0.0006 ± 0.007$, $t = 0.085$, df = 51, $P = 0.933$, adjusted $R^2 = -0.019$; difference in adjusted TEE: estimate ± SE: $0.002 ± 0.007$, $t = 0.312$, df = 51, $P = 0.756$, adjusted $R^2 = -0.017$; Supplementary Fig. 2d).

Change in adjusted TEE was negatively correlated with change in FM and positively with change in FFM within an individual (Supplementary Note 2 and Supplementary Fig. 3a–d), when using the change in FFM (kg/week), FM (kg/week), and TEE (MJ/

week) between the two repeated TEE measurements in alternative analyses. We found the same results when restricting the analyses to a subset of individuals ($N = 53$) for which the interval between repeated TEE measurements was longer than 4 weeks, (Supplementary Note 2 and Supplementary Fig. 3e–g).

## Discussion

Our findings show that TEE measurements are repeatable in adults, also in adults older than 50 y, and over extended periods of time. The stability in adjusted TEE among adults is remarkable given the degree to which body weight and composition changed among subjects in our sample.

The repeatability estimate $R$ (also ICC) for TEE adjusted for FFM, FM, sex, and age was high for adults (0.64), and in the range of previously reported estimates for mass-adjusted metabolic rates (range: 0.5–0.7) of small mammals[19–21,24,34,35]. Within adult humans, some individuals exhibit consistently high TEE for their body size and composition, while others are consistently low. These metabolic profiles are durable over 8 + y (Supplementary Fig. 1a-c) and, as far as can be determined here, throughout adulthood.

The environmental, genetic, and behavioral contributions to maintaining high or low adjusted TEE remain unclear. While we lack independent measures of physical activity for subjects in this dataset, changes in FFM suggest exercise has a relatively modest effect on the magnitude or maintenance of adjusted TEE. For adults 20–60 y, changes in FFM, an indirect proxy for strenuous physical activity, were positively associated with changes in adjusted TEE. However, there are numerous forms of physical activity that do not lead to changes in FFM and thus, objective measures of physical activity are needed to further examine the effects of behavior on the maintenance and repeatability of adjusted TEE.

Typically, repeatability sets an upper limit to heritability[36,37], and thus the high repeatability of adult TEE may indicate some degree of heritability and genetic influence. A sibling study in humans (37 siblings aged 5–9 y) reported a low heritability (h2 = 0.11) of TEE adjusted for resting metabolic rate[38]. Alternatively, TEE could exhibit considerable developmental plasticity, but remain stable in adulthood after the closure of some critical developmental window. For example, Pontzer[39,40] has argued that adjusted TEE is largely constrained and under homeostatic control, and that adjusted TEE develops during childhood and adolescence in response to environmental cues regarding activity demands and food availability. Plasticity in TEE through childhood would be in line with the low degree of repeatability for children in this study. More work is needed to investigate the ontogeny of metabolic physiology, variability in adjusted TEE throughout childhood, and the establishment of high versus low adjusted TEE in adults.

The multi-response models showed that TEE and body fat (both FM and body fat percentage) were not correlated at the within- or among-individual level. Models using the entire dataset of adults 20–60 y ($N = 267$ subjects) and those using a subset of individuals ($N = 53$ subjects) for which the time between measurements exceeded 4 weeks showed the same results. These results are further strengthened by our analyses of the relationship between changes in body composition and weight and TEE, after accounting for its covariation with FFM, FM, age, and sex. If greater TEE was protective against gaining fat, then subjects with greater adjusted TEE, or positive changes in adjusted TEE, should have experienced less weight and fat gain. Instead, we found a positive relationship between the difference in adjusted TEE between measurements and change in body weight (in both datasets), and no relationship between any measure of adjusted TEE (time 1, difference between measures, or average) and

change in body fat percentage (in both datasets) among adults in our sample. The similarity between the results in both datasets indicates that these findings are not an artefact of measurement error or short time intervals. It is noteworthy that there was a trend towards a negative relationship between adjusted TEE1 and change in body weight ($P = 0.094$) in the subset of individuals for which the period between repeated TEE measurements was longer than 4 weeks, but none of the other relationships indicated that TEE is associated with changes in body weight or composition in this subset. Therefore, we cannot rule out the possibility that TEE would be associated with small changes in body weight over much longer timeframes. But the results of the current study are consistent with those of previous work conducted on adults[13,14] and children[9–12] which reported no relationship between TEE and change in body fat percentage.

Our analyses of weight change with TEE change (Fig. 2c) are particularly informative regarding the relationship between expenditure and weight change. In a simple model in which changes in TEE resulted directly in equivalent changes in weight, we would expect to observe a ratio of ~7 MJ per kg[41]; every additional ~7 MJ burned would reduce weight by 1 kg. Adult TEE is ~10 MJ/d, and thus every 10% change represents 1 MJ/d or 7 MJ/week. This simple model would predict a negative slope of −0.1 for Fig. 2c: subjects who increase TEE by 10% should experience weight loss at ~1 kg/week, while those who decrease TEE by 10% should experience a ~1 kg/week weight gain. This modeling approach is a limited first-approximation because we do not know the time course of TEE change between measurements and the ratio of MJ/kg will vary with different tissues. Nonetheless, the observed relationship between TEE and weight changes, which is positive in our sample, clearly challenges the expectation that decreased TEE is associated with weight gain.

This study is limited by the lack of additional physiological or behavioral measures. Objective measures of physical activity would enable us to examine the contribution of changes in daily activity to observed changes in TEE and body composition. Similarly, measures of organ size or resting expenditure would enable us to investigate whether age-related changes in organ size and activity affect repeatability in TEE, particularly among children. As the DLW database expands, the addition of more associated variables such as organ size and activity will strengthen its utility.

Humans, like other species, exhibit a substantial degree of variation in TEE. Even after accounting for the effects of physical activity and anthropometric variables such as FFM, individuals can still vary by 20% or more[42]. Our analyses here show that having a "fast" or "slow" metabolism is a repeatable, durable trait for adults that is consistent over years. However, we find no evidence that subjects with lower adjusted TEE are at increased risk of gaining body fat, nor that higher adjusted TEE protects against weight gain. The causes and consequences of metabolic variation in humans remain a critical focus for future investigation.

## Methods

**Criteria for inclusion of individuals from the IAEA DLW database.** We included all individuals in the IAEA DLW database (https://doubly-labelled-water-database.iaea.org/ home) which had their TEE measured at least twice and that were at least 1 year old. Only healthy individuals and those not involved in athletic competition or training were included. This resulted in 696 TEE measurements of 348 healthy adults and 114 TEE measurements of 47 children (22 girls and 25 boys). These repeated TEE measurements were collected from 225 women (age range at the first TEE measurement: 21–81) and 123 men (age range at the first TEE measurement: 22–82). No repeated TEE measurements were available between the ages of 7–20 years. For children, 32 measurements were collected at age 2, 41 measurements at age 4, and 41 measurements at age 6). For 20 children, three repeated measurements were available. We excluded TEE measurements from children younger than 1 year because the relationship between TEE and FFM in this age group appears to differ from that for older children[11]. Mean time interval between two TEE measurements of the same individual was 1.9 ± 2.88 y

(range: 0.04–8.2 y). 66.0% were repeated TEE within 1 y, 14.2% were repeated 2–4 y after initial TEE measurement, and 19.8% were repeated 5–8 y after initial TEE measurement. All measurements (body weight, TEE, FFM, and FM) were obtained directly from the IAEA DLW database. Dilution space (a measure of the total body water pool) and FFM hydration (water content of FFM) are an integral part of the determination of body composition and TEE, and we corrected measurements for age- and body weight-related variation in dilution space and FFM hydration. FFM measurements included in this study were estimated via the isotope dilution method, and FM and body fat percentage were determined by subtracting FFM from body weight.

All of the studies that provided data into the IAEA DLW database were locally ethically reviewed and approved. The present paper is based on a secondary analysis of these compiled data and such analyses do not require ethical permission.

**Repeatability estimate 'R'.** We estimated repeatability $R$ as $R = V_G/(V_G + V_R)$, where $V_G$ = inter-individual variance and $V_R$ = intra-individual variance[30]. The estimate $R$ is more generally also referred to as the intra-class correlation (ICC). This estimate has previously been used to estimate the reproducibility of metabolic rate measurements in animals[18,34,43,44].

## Statistical analysis

*Repeatability estimate R.* All analyses were conducted using $R$ version 3.6.2[45]. Repeatabilities are tested at the boundary and tests are therefore typically one-tailed, and all other tests are two-tailed. We estimated repeatability $R$ using a mixed effects model framework in the R package rptR[30]. We included FFM, FM, sex, and age as fixed factors. Controlling for these fixed effects removes the influence of their variance from the estimate $R$[30]. Thus, we estimated the adjusted repeatability of TEE, after controlling for the body composition variables, age, and sex as sources of variation in the dataset. We ln-transformed the response variable TEE and the fixed factors FFM and FM, and used individual ID as random factor. In a first step, we estimated $R$ of TEE for all individuals older > 1 y ($N = 395$ individuals) together. But because the relationship between FFM and TEE may change in children as their levels of physical activity and the relative size of metabolically active organs change during development, we predicted $R$ of TEE of children would be lower than that of adults. Thus, we estimated $R$ of TEE also for children (2–6 y; $N = 47$ individuals) and adults (21–89 y; $N = 348$ individuals) separately. We also assessed whether body mass, after controlling for age and sex, was repeatable for all individuals together, and adults and children separately.

We estimated confidence intervals for repeatabilities by parametric bootstrapping. We used likelihood ratio tests (LRT) and permutation tests to test for statistical significance against the null hypothesis that TEE is not repeatable. LRTs compare the fit of the model including the grouping factor of interest (here individual ID) and one excluding that factor, which constrains the group-level variance to zero. Permutation of residuals randomizes the grouping factor against the response variable, followed by refitting the model to the randomized data. $R$ varies from 0 to 1 and we considered TEE to be repeatable if the 95% confidence interval around $R$ did not include zero.

*Is TEE associated with short-term changes in weight or body composition?* We restricted this analysis to a subset of individuals aged 20–60 y ($N = 267$). We excluded children from this analysis due to their continued somatic growth and expected increase in TEE during aging[11], and we excluded subjects older than 60 y because TEE, FFM and FM are all known to decrease at this age[46,47]. We used two approaches to address this question.

In a first approach, we evaluated covariations among TEE and FM (Model 1) and among TEE and body fat percentage (Model 2) by fitting two multivariate Bayesian mixed models using the package MCMCglmm[48], including individual ID as random factor. We included sex, age and FFM as fixed factors. We fitted the fixed effect FFM only for the trait TEE because FFM is the main predictor of TEE (see Supplementary Table 1b) and also known to account for a large proportion of between-individual variation in TEE[49–51]. We repeated this analysis using a subset of subjects for which the time between measurements exceeded 4 week ($N = 53$) to evaluate covariations among TEE and FM (Model 3) and among TEE and body fat percentage (Model 4). We partitioned phenotypic variances and covariances into within- and between-individual components[32] using an unstructured variance–covariance matrix[48]. We calculated correlations between traits at the phenotypic ($r_p$), among-individual ($r_{ind}$), and within-individual ($r_e$) level. All continuous variables were standardized to a mean of 0 and a variance of 1. We estimated the correlation between both traits by comparing the variance–covariance divided by the square root of the product of variances[52]. We used inverse gamma priors, and MCMC sampling scheme of 900,000 total iterations with a 30,000 iteration burn-in and sampling (thinning) interval of 250. This yielded Monte Carlo Markov Chains with a sample size of 3480. We estimated the level of non-independence between successive samples in the chain using the 'autocorr' function in the coda package[53]. For all models, we ran three independent chains and assessed MCMC convergence and mixing visually by plotting the traces and densities of sampled values across iterations, and confirmed convergence using the Gelman-Rubin convergence criterion (all < 1.1) using the coda package[53].

In a second approach, we calculated a body size- and composition-adjusted TEE, which accounts for the covariation of TEE with body size and composition. We calculated an adjusted TEE for each subject at each time point based on a multiple regression model with TEE as the dependent variable and FFM, FM, age, and sex as independent variables (Supplementary Table 1). We ln-transformed TEE, FFM, and FM for these models. Using the predicted TEE and observed TEE for each measurement, we calculated adjusted TEE as

$$\text{adjusted TEE} = (\text{Observed TEE}/\text{predicted TEE}) \times 100$$

An adjusted TEE of 120% indicates a measured TEE that is 20% greater than predicted from body composition variables; an adjusted TEE of 80% is 20% less than predicted, and so on. We tested whether adjusted TEE1, the mean of adjusted TEE1 and adjusted TEE2, or the difference between adjusted TEE1 and adjusted TEE2 (i.e., adjusted TEE2 – adjusted TEE1) were associated with changes in body weight and body fat percentage using linear models. Moreover, we tested whether the change in adjusted TEE per week was associated with future changes in body weight and body fat percentage using linear models.

We analyzed the full 20–60 y dataset ($N = 267$) as well as the subset of subjects for which the time between measurements exceeded 4 week ($N = 53$). The latter analysis of longer-duration measurements reduces the potential effect of measurement error on the calculated rate of FFM and FM change. Short-term variations in body weight are mainly due to changes in FFM and to a smaller extent due to changes in FM[54]. Small errors in the determination of FFM, either due to measurement imprecision, variation in the hydration of FFM (used to calculate FFM from isotope dilution), will lead to errors when determining the rate of change (kg/wk) for both FFM and FM. Short time intervals between repeated TEE measurements will inflate these errors, exaggerating the calculated rate of change. Restricting our analysis to subjects for which the interval between repeated TEE measurements was longer than 4 weeks reduces this effect.

**Reporting summary**. Further information on research design is available in the Nature Research Reporting Summary linked to this article.

## Data availability

All data supporting the analyses and results in this paper are available from the Doubly Labeled Water Database (https://doubly-labelled-water-database.iaea.org/home, https://www.dlwdatabase.org/) upon reasonable request. Because of human study participant confidentiality the database is not open access. However, access to components of the data is freely available to perform novel and approved analyses. Details of the application process are available at https://doubly-labelled-water-database.iaea.org/dataAnalysisInstructions. Data published in this paper will be provided normally within 3 weeks of receipt of the request. Such data has unrestricted use except we ask users not to share the data with others or post it on social media or other internet sites. If users wish to publish analyses of such provided data, we ask that they adhere to the procedures established to ensure fair credit for those contributing the data into the DLW database.

## Code availability

We provide the source code used to perform analysis and output files through the OSF repository (https://osf.io/6q2kz/).

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

## Acknowledgements

The DLW database, which can be found at https://doubly-labelled-water-database.iaea.org/home or https://www.dlwdatabase.org/, is generously supported by the IAEA, Taiyo Nippon Sanso and, SERCON. We are grateful to David Ludwig and Cara Ebbeling for their contributed data. We are grateful to these companies for their support and especially to Takashi Oono for his tremendous efforts at fund raising on our behalf. The authors also gratefully acknowledge funding from the US National Science Foundation (BCS-1824466) awarded to Herman Pontzer.

## Author contributions

Y.Y., H.S., P.N.A., L.F.A., LJ.A., L.A., I.B., K.B.-A., E.E.B., S.B., A.G.B., C.V.C.B., P.B., M.S.B., N.F.B., S.G.J.A.C., G.L.C., J.A.C., S.K.D., L.R.D., U.E., S.E., T.F., B.W.F., A.H.G., M.G., C.H., A.E.H., M.B.H., S.H., N.J., A.M.J., P.K., K.P.K., M.K., W.E.K., R.F.K., E.V.L., W.R.L., N.L., C.K.M., A.C.M., E.P.M., J.C.M., J.P.M., M.L.N., T.A.N., R.M.O., K.H.P., Y.P.P., J.P-R., G.P., R.L.P., R.A.R., S.B.R., D.A.R., E.R., R.M.R., S.B.R., A.J.S., A.M.S., E.S., S.S.U., G.V., L.M.V.E., E.A.V.M., J.C.K.W., G.W., B.M.W., J.Y., T.Y., X.Z., A.J.M.-A., C.U.L., A.H.L., J.R., D.A.S., K.R.W., W.W.W., J.R.S., and H.P. contributed data. R.R. and H.P. designed the study. R.R. performed the analyses, with support from H.P. on analytical approaches. R.R. and H.P. led the writing with input from J.R.S., D.A.S., W.W.W., K.R.W., H.S., and Y.Y.

## Competing interests

## Additional information

Rebecca Rimbach [1,2,65], Yosuke Yamada [3,4], Hiroyuki Sagayama [5], Philip N. Ainslie [6], Lene F. Anderson [7], Liam J. Anderson [6,8], Lenore Arab [9], Issaad Baddou [10], Kweku Bedu-Addo [11], Ellen E. Blaak [12], Stephane Blanc [13,14], Alberto G. Bonomi [15], Carlijn V. C. Bouten [16], Pascal Bovet [17], Maciej S. Buchowski [18], Nancy F. Butte [19], Stefan G. J. A. Camps [12], Graeme L. Close [6], Jamie A. Cooper [13], Sai Krupa Das [20], Lara R. Dugas [21], Ulf Ekelund [22], Sonja Entringer [23,24], Terrence Forrester [25], Barry W. Fudge [26], Annelies H. Goris [12], Michael Gurven [27], Catherine Hambly [28], Asmaa El Hamdouchi [10], Marije B. Hoos [12], Sumei Hu [29], Noorjehan Joonas [30], Annemiek M. Joosen [12], Peter Katzmarzyk [31], Kitty P. Kempen [12], Misaka Kimura [3], William E. Kraus [32], Robert F. Kushner [33], Estelle V. Lambert [34], William R. Leonard [35], Nader Lessan [36], Corby K. Martin [31], Anine C. Medin [7,37], Erwin P. Meijer [12], James C. Morehen [6,38], James P. Morton [6], Marian L. Neuhouser [39], Theresa A. Nicklas [19], Robert M. Ojiambo [40,41], Kirsi H. Pietiläinen [42], Yannis P. Pitsiladis [43], Jacob Plange-Rhule [11,66], Guy Plasqui [44], Ross L. Prentice [39], Roberto A. Rabinovich [45],

Susan B. Racette [46], David A. Raichlen[47], Eric Ravussin [31], Rebecca M. Reynolds[48], Susan B. Roberts [20], Albertine J. Schuit[49], Anders M. Sjödin [50], Eric Stice[51], Samuel S. Urlacher[52], Giulio Valenti[12], Ludo M. Van Etten[12], Edgar A. Van Mil[53], Jonathan C. K. Wells [54], George Wilson[6], Brian M. Wood[55,56], Jack Yanovski [57], Tsukasa Yoshida [5], Xueying Zhang[28,29], Alexia J. Murphy-Alford[58], Cornelia U. Loechl[58], Amy H. Luke [59✉], Jennifer Rood [31✉], Dale A. Schoeller [60✉], Klaas R. Westerterp [61✉], William W. Wong [19✉], John R. Speakman [28,29,62,63,65✉], Herman Pontzer [1,64,65✉] & the IAEA DLW Database Consortium*

[1]Evolutionary Anthropology, Duke University, Durham, NC, USA. [2]School of Animal, Plant & Environmental Sciences, University of the Witwatersrand, Johannesburg, South Africa. [3]National Institute of Health and Nutrition, National Institutes of Biomedical Innovation, Health and Nutrition, Tokyo, Japan. [4]Institute for Active Health, Kyoto University of Advanced Science, Kyoto, Japan. [5]Faculty of Health and Sport Sciences, University of Tsukuba, Ibaraki, Japan. [6]Research Institute for Sport and Exercise Sciences, Liverpool John Moores University, Liverpool, UK. [7]Department of Nutrition, Institute of Basic Medical Sciences, University of Oslo, 0317 Oslo, Norway. [8]Crewe Alexandra Football Club, Crewe, UK. [9]David Geffen School of Medicine, University of California, Los Angeles, USA. [10]Unité Mixte de Recherche en Nutrition et Alimentation, CNESTEN-Université Ibn Tofail URAC39, Regional Designated Center of Nutrition Associated with AFRA/IAEA, Rabat, Morocco. [11]Department of Physiology, Kwame Nkrumah University of Science and Technology, Kumasi, Ghana. [12]Maastricht University, Maastricht, The Netherlands. [13]Nutritional Sciences, University of Wisconsin, Madison, WI, USA. [14]Institut Pluridisciplinaire Hubert Curien, CNRS Université de Strasbourg, Strasbourg UMR7178, France. [15]Phillips Research, Eindhoven, The Netherlands. [16]Department of Biomedical Engineering and Institute for Complex Molecular Systems Eindhoven Unversity of Technology, Eindhoven, The Netherlands. [17]Institute of Social and Preventive Medicine, Lausanne University Hospital, Lausanne, Switzerland. [18]Division of Gastroenterology, Hepatology and Nutrition, Department of Medicine, Vanderbilt University, Nashville, Tennessee, USA. [19]Department of Pediatrics, Baylor College of Medicine, USDA/ARS Children's Nutrition Research Center, Houston, Texas, USA. [20]Jean Mayer USDA Human Nutrition Research Center on Aging, Tufts University, 711 Washington St, Boston, MA 02111, USA. [21]Department of Public Health Sciences, Parkinson School of Health Sciences and Public Health, Loyola University, Maywood, IL, USA. [22]Department of Sport Medicine, Norwegian School of Sport Sciences, Oslo, Norway. [23]Charité – Universitätsmedizin Berlin, corporate member of Freie Universität Berlin, Humboldt-Universität zu Berlin, and Berlin Institute of Health (BIH), Institute of Medical Psychology, Berlin, Germany. [24]University of California Irvine, Irvine, California, USA. [25]Solutions for Developing Countries, University of the West Indies, Mona, Kingston, Jamaica. [26]University of Glasgow, Glasgow, UK. [27]Department of Anthropology, University of California Santa Barbara, Santa Barbara, CA, USA. [28]Institute of Biological and Environmental Sciences, University of Aberdeen, Aberdeen, UK. [29]State Key Laboratory of Molecular Developmental Biology, Institute of Genetics and Developmental Biology, Chinese Academy of Sciences, Beijing, China. [30]Central Health Laboratory, Ministry of Health and Wellness, Port Louis, Mauritius. [31]Pennington Biomedical Research Center, Baton Rouge, Louisiana, USA. [32]Department of Medicine, Duke University, Durham, North Carolina, USA. [33]Northwestern University, Chicago, IL, USA. [34]Research Unit for Exercise Science and Sports Medicine, University of Cape Town, Cape Town, South Africa. [35]Department of Anthropology, Northwestern University, Evanston, IL, USA. [36]Imperial College London Diabetes Centre, Imperial College London, London, UK. [37]Department of Nutrition and Public Health, Faculty of Health and Sport Sciences, University of Agder, 4630 Kristiansand, Norway. [38]The FA Group, Burton-Upon-Trent, Staffordshire, UK. [39]Division of Public Health Sciences, Fred Hutchinson Cancer Research Center and School of Public Health, University of Washington, Seattle, WA, USA. [40]Moi University, Eldoret, Kenya. [41]University of Global Health Equity, Kigali, Rwanda. [42]Helsinki University Central Hospital, Helsinki, Finland. [43]University of Brighton, Eastbourne, UK. [44]Department of Nutrition and Movement Sciences, Maastricht University, Maastricht, The Netherlands. [45]University of Edinburgh, Edinburgh, UK. [46]Program in Physical Therapy and Department of Medicine, Washington University School of Medicine, St. Louis, Missouri, USA. [47]Biological Sciences and Anthropology, University of Southern California, California, USA. [48]Centre for Cardiovascular Sciences, Queen's Medical Research Institute, University of Edinburgh, Edinburgh, UK. [49]University of Tilburg, Tilburg, The Netherlands. [50]Department of Nutrition, Exercise and Sports, Copenhagen University, Copenhagen, Denmark. [51]Stanford University, Stanford, CA, USA. [52]Department of Anthropology, Baylor University, Waco, TX, USA. [53]Maastricht University, Maastricht and Lifestyle Medicine Center for Children, Jeroen Bosch Hospital's-Hertogenbosch, 's-Hertogenbosch, The Netherlands. [54]Population, Policy and Practice Research and Teaching Department, UCL Great Ormond Street Institute of Child Health, London, UK. [55]University of California Los Angeles, Los Angeles, USA. [56]Max Planck Institute for Evolutionary Anthropology, Department of Human Behavior, Ecology, and Culture, Leipzig, Germany. [57]Growth and Obesity, Division of Intramural Research, NIH, Bethesda, MD, USA. [58]Nutritional and Health Related Environmental Studies Section, Division of Human Health, International Atomic Energy Agency, Vienna, Austria. [59]Division of Epidemiology, Department of Public Health Sciences, Loyola University School of Medicine, Maywood, Illinois, USA. [60]Biotech Center and Nutritional Sciences University of Wisconsin, Madison, Wisconsin, USA. [61]School of Nutrition and Translational Research in Metabolism, University of Maastricht, Maastricht, The Netherlands. [62]Center for Energy Metabolism and Reproduction, Shenzhen Institutes of Advanced Technology, Chinese Academy of Sciences, Shenzhen, China. [63]CAS Center of Excellence in Animal Evolution and Genetics, Kunming, China. [64]Duke Global Health Institute, Duke University, Durham, NC, USA. [65]These authors contributed equally: Rebecca Rimbach, John R. Speakman, Herman Pontzer. [66]Deceased: Jacob Plange-Rhule. *A list of authors and their affiliations appears at the end of the paper. ✉email: rrimbach@gmail.com; yyamada831@gmail.com; sagayama.hiroyuki.ka@u.tsukuba.ac.jp; aluke@luc.edu; jennifer.rood@pbrc.edu; dschoell@nutrisci.wisc.edu; k.westerterp@maastrichtuniversity.nl; wwong@bcm.edu; j.speakman@abdn.ac.uk; herman.pontzer@duke.edu

## The IAEA DLW Database Consortium

John R. Speakman[28,29,62,63]

A full list of members and their affiliations appears in the Supplementary Information.

