## [Peer Review File · Nature Communications]

Reviewers' Comments:

Reviewer #1:

Remarks to the Author:

The authors successfully addressed all my minor points from the previous submission. I do not have any additional comments and requests for clarifications.

Reviewer #2:

Remarks to the Author:

The manuscript reports the repeatability of human total energy expenditure (TEE, quantified by doubly labeled water) over a period of a few weeks to a few years. The results show a clear contrast between rather high repeatability in adults and very low (statistically non-detectable) repeatability in 1-6 year-old children. Furthermore, the study reports correlations of changes of TEE with changes in body composition. Results from the latter analysis are largely non-significant, suggesting that the TEE does not causally affect changes in body composition.

This is a revised version of a manuscript that I had reviewed before. Improvements from the previous revision are visible, but overall rather minor. I still find that the manuscript makes a tough read, mostly because there is largely now guidance through the long results section. Some of the analyses are redundant it is quite hard to make out what they are done for. I had previously asked for more context. A bit has been done, but I think there is still room for significant improvements.

The manuscript states that the analysis confirms Wong et al. (2014). However, my hunch is if the data from Wong et al. (2014) are included in the current analysis. If this is true, I wouldn't consider this an independent confirmation. A clear statement on data overlap is needed.

My main concern is section "(b) Repeatability using adjusted TEE". In essence I think this can be removed without loss and for the sake of improved readability. The first half of the section presents a two-step analysis that simply repeats the adjusted repeatability analysis from section (a) in an inferior way. The second half of the section, tests for average differences between first and second measurements. I don't understand the motivation for doing so, since values have been corrected for age etc., so effects are expectedly non-significant. The relevant effect has been adjusted for, so why is there an expectation for a difference?

The first part of results section (c) reports on correlations between measures of body composition. I would expect that this is long known (since the data I would think are comparatively cheap to acquire). Furthermore, the correlations are also covered by the covariance decomposition analysis. I suggest to focus the presentation on associations with TEE, which is the main topic of the manuscript. This would again improve readability.

The covariance decomposition by a multi-response model analysis is interesting in principle, but the manuscript does not really discuss the implications. What does it mean if correlations differ at the between- vs. within-individual level? 95% CI for the covariance decomposition presented in Table 1 look suspiciously small given the sample sizes. They might well be correct (since they depend partly on the correlation structure in the data), but I would be more convinced if the analysis code was available.

I still find the length of the author list inappropriate. There are 810 data points and 82 authors. The acknowledgement sections states 8 authors that contributed to conception, analysis and writing. The remaining authors contributed on average 11 data points (5-6 individuals tested

twice). This average contribution is not substantial in my view (removing even a few dozen data points wouldn't change the conclusion). The authors argue that the production of the data is challenging and costly. But these data were clearly not produced for the purpose of the analysis (otherwise I would consider this a very poor sampling design). I find it problematic if the intellectual input from reviewers is larger than that of 90% of the authors. My suggestion is to list the eight authors that contributed to the manuscript and list the rest as members of the IAEA DLW database group (that appears as an author anyway).

Minor comments

I would suggest to write adjusted TEE with lower case "a"

L221: Suggest "we used multi-response models to examine" ("multiple regression" typically means multiple covariates).

L232: The degrees of freedom are incomplete. F tests have numerator and denominator degrees of freedom.

L248: It is hard to understand "predicted TEE" without reading the methods first.

L502 + 533: It is unnecessary to exclude the fixed factor of age in the analysis. The age-FFM correlation just affects the variance composition between age and FFM. The model is a little better with age included – and it also makes the analysis consistent with the analysis in adults.

L524: What is the motivation for forming this ratio instead of just using residuals?

L392: It would be useful to report the heritability estimate more precisely than just "moderate".

Effect size statistics like Cohen's d should include SEs.

Reviewer #3:

Remarks to the Author:

the authors have responded satisfactorily to the issues that were raised in the previous review of the manuscript.

Reviewer #4:

Remarks to the Author:

In my previous review I commented on how much of the data that underpinned the conclusion that TEE did not predict change in weight was from people who had a very short duration of follow up (<4 weeks). I also pointed out that in the small sub-group of people for whom the time between repeats was longer, there was a negative relationship with change in weight and a p value that was suggestive.

The authors have responded to this was focusing their attention on why one should make too much of the p value. With respect, I don't think that they have given sufficiently serious attention to the major point that was being made which was that they have predominantly studied short term change in weight. I suggested that this should be acknowledged but they have ignored this. It is noteworthy at least that the results are different in the small group of people with longer follow up.

The point here is that the take home message that some people will see is that TEE isn't related to weight gain. This has important potential public health implications if true. However most people reading that conclusion will assume it is based on rather longer follow up than less than 4 weeks. The authors cannot conclude that people with higher TEE are more or less likely to gain weight over the longer term because they have so little data beyond 4 weeks. The data that they do have

does tend to undermine their own conclusion.

Thus it seems to me to be important to make it clear in the title that TEE is not predictive of "short term" changes in body composition. It would also be necessary to make this much clearer in the conclusions.

REVIEWER COMMENTS

Reviewer #1 (Remarks to the Author):

The authors successfully addressed all my minor points from the previous submission. I do not have any additional comments and requests for clarifications.

Reviewer #2 (Remarks to the Author):

The manuscript reports the repeatability of human total energy expenditure (TEE, quantified by doubly labeled water) over a period of a few weeks to a few years. The results show a clear contrast between rather high repeatability in adults and very low (statistically non-detectable) repeatability in 1-6 year-old children. Furthermore, the study reports correlations of changes of TEE with changes in body composition. Results from the latter analysis are largely non-significant, suggesting that the TEE does not causally affect changes in body composition.

This is a revised version of a manuscript that I had reviewed before. Improvements from the previous revision are visible, but overall rather minor. I still find that the manuscript makes a tough read, mostly because there is largely now guidance through the long results section. Some of the analyses are redundant it is quite hard to make out what they are done for. I had previously asked for more context. A bit has been done, but I think there is still room for significant improvements.

Response: We followed this advice and the comments detailed below. We reduced redundant analyses and provide more context for each included analysis. The analyses identified as redundant by the reviewer have been moved to Supplementary material to provide interested readers with the additional results, all of which confirm the primary results reported in the main text.

The manuscript states that the analysis confirms Wong et al. (2014). However, my hunch is if the data from Wong et al. (2014) are included in the current analysis. If this is true, I wouldn't consider this an independent confirmation. A clear statement on data overlap is needed.

Response: We deleted this sentence and revised the first paragraph of the Conclusions section: "Our findings show that TEE measurements are repeatable in adults, also in adults older than 50 y, and over extended periods of time. The stability in adjusted TEE among adults is remarkable given the degree to which body weight and composition changed among subjects in our sample." (lines 359-363)

My main concern is section "(b) Repeatability using adjusted TEE". In essence I think this can be removed without loss and for the sake of improved readability. The first half of the section presents a two-step analysis that simply repeats the adjusted repeatability analysis from section (a) in an inferior way. The second half of the section, tests for average differences between first and second measurements. I don't understand the motivation for doing so, since values have been corrected for age

etc., so effects are expectedly non-significant. The relevant effect has been adjusted for, so why is there an expectation for a difference?

Response: We removed this section from the manuscript and moved it to the supplementary materials. We test for average differences between the first and the second measurement because even after accounting for variation in FFM, FM, age and sex, there is still inter-individual variation in TEE. Therefore, we think that this analysis has its merit. But since this part is redundant with section a), we moved it to the supplementary material to improve readability of the manuscript.

The first part of results section (c) reports on correlations between measures of body composition. I would expect that this is long known (since the data I would think are comparatively cheap to acquire). Furthermore, the correlations are also covered by the covariance decomposition analysis. I suggest to focus the presentation on associations with TEE, which is the main topic of the manuscript. This would again improve readability.

Response: We followed this suggestion and removed this part from the manuscript and moved it to the supplementary materials.

The covariance decomposition by a multi-response model analysis is interesting in principle, but the manuscript does not really discuss the implications. What does it mean if correlations differ at the between- vs. within-individual level? 95% CI for the covariance decomposition presented in Table 1 look suspiciously small given the sample sizes. They might well be correct (since they depend partly on the correlation structure in the data), but I would be more convinced if the analysis code was available.

Response: We have revised this part of the data analysis and results section.

We introduce the difference between among- and within-individual correlation in the introduction: "Firstly, we used a multi-response model to decompose the covariance between TEE and FM on a between (r_{ind}) and within-individual (r_e) level, where r_{ind} indicates whether individual mean values of traits correlate, and where r_e indicates when the change in one trait between two time points is correlated with the change in another trait over the same period within an individual^{32,33} (see Methods). Thus, r_e represents combined, reversible changes in traits that occur within an individual, and r_{ind} reflects genetic and permanent environmental effects that are responsible for the association between the traits^{32,33}." (lines 225-233)

We also revised the analysis section. We increased the number of iterations, iteration burn-in and sampling (thinning) interval, and we provide the R code:

"In a first approach, we evaluated covariations among TEE and FM (Model 1) and among TEE and body fat percentage (Model 2) by fitting two multivariate Bayesian mixed models using the package MCMCglmm⁴⁸, including individual ID as random factor. We included sex, age and FFM as fixed factors. We estimated the fixed effect FFM only for the trait TEE because FFM is the main predictor of TEE (see Supplementary Table 1B) and also known to account for a large proportion of between-individual variation in TEE⁴⁹⁻⁵¹. We repeated this analysis using a subset of subjects for which the time between measurements exceeded 4 wk (N = 53) to evaluate covariations among TEE and FM (Model 3) and among TEE and body fat percentage (Model 4). We partitioned phenotypic variances and covariances into within- and between-individual components⁵² using an unstructured variance-

covariance matrix⁴⁸. We calculated correlations between traits at the phenotypic (r_p), among-individual (r_{ind}), and within-individual (r_e) level. All continuous variables were standardized to a mean of 0 and a variance of 1. We estimated the correlation between both traits by comparing the variance-covariance divided by the square root of the product of variances⁵³. We used inverse gamma priors, and MCMC sampling scheme of 900000 total iterations with a 30000 iteration burn-in and sampling (thinning) interval of 250. This yielded Monte Carlo Markov Chains with a sample size of 3480. We estimated the level of non-independence between successive samples in the chain using the 'autocorr' function in the coda package⁵⁴. For all models, we ran three independent chains and assessed MCMC convergence and mixing visually by plotting the traces and densities of sampled values across iterations, and confirmed convergence using the Gelman-Rubin convergence criterion (all <1.1) using the coda package⁵⁴." (lines 544-566)

I still find the length of the author list inappropriate. There are 810 data points and 82 authors. The acknowledgement sections states 8 authors that contributed to conception, analysis and writing. The remaining authors contributed on average 11 data points (5-6 individuals tested twice). This average contribution is not substantial in my view (removing even a few dozen data points wouldn't change the conclusion). The authors argue that the production of the data is challenging and costly. But these data were clearly not produced for the purpose of the analysis (otherwise I would consider this a very poor sampling design). I find it problematic if the intellectual input from reviewers is larger than that of 90% of the authors. My suggestion is to list the eight authors that contributed to the manuscript and list the rest as members of the IAEA DLW database group (that appears as an author anyway).

Response: We understand the Reviewer's perspective, but we believe co-authorship of those listed is appropriate. Each of them contributed data to the database and therefore have contributed to this analysis. A promise made to those supplying their data to the database is co-authorship on papers that use the data. We cannot breach this promise on the basis of comments of a single reviewer.

Minor comments

I would suggest to write adjusted TEE with lower case "a"

Response: We followed this suggestion and changed "Adjusted" to "adjusted" throughout the manuscript.

L221: Suggest "we used multi-response models to examine" ("multiple regression" typically means multiple covariates).

Response: We followed this suggestion and changed the text accordingly.

L232: The degrees of freedom are incomplete. F tests have numerator and denominator degrees of freedom.

Response: We deleted this analysis when reducing the number of analyses to increase readability.

L248: It is hard to understand "predicted TEE" without reading the methods first.

Response: We removed section b) from the main text and moved the section to supplementary materials. In the supplementary materials, we included an additional sentence to clarify what “predicted TEE” is:

“We computed a predicted TEE from a multiple regression model with TEE as the dependent variable and FFM, FM, age, and sex as independent variables.”

L502 + 533: It is unnecessary to exclude the fixed factor of age in the analysis. The age-FFM correlation just affects the variance composition between age and FFM. The model is a little better with age included – and it also makes the analysis consistent with the analysis in adults.

Response: We followed this advise and included age as a fixed factor in the analyses.

L524: What is the motivation for forming this ratio instead of just using residuals?

Response: The ratio and residuals will give the same results but we think that the ratio is more intuitive and easier for interpretation than residual. The ratio allows the reader to understand what a higher or lower than predicted TEE is and since a low TEE has been hypothesized to be a risk factor for obesity, we think using this ratio is useful in this context.

L392: It would be useful to report the heritability estimate more precisely than just “moderate”.

Response: We followed this suggestion and revised the sentence:

“A sibling study in humans (37 siblings aged 5 – 9 y) reported a low heritability (0.11) of TEE adjusted for resting metabolic rate.” (lines 379-380)

Effect size statistics like Cohen’s d should include SEs.

Response: We moved this section to the supplementary results. We could not find out how to calculate the SEs for Cohen’s d, but instead we report 95% CIs for Cohen’s d.

Reviewer #3 (Remarks to the Author):

the authors have responded satisfactorily to the issues that were raised in the previous review of the manuscript.

Reviewer #4 (Remarks to the Author):

In my previous review I commented on how much of the data that underpinned the conclusion that TEE did not predict change in weight was from people who had a very short duration of follow up (<4 weeks). I also pointed out that in the small sub-group of people for whom the time between repeats was longer, there was a negative relationship with change in weight and a p value that was suggestive.

The authors have responded to this was focusing their attention on why one should make too much of the p value. With respect, I don't think that they have given sufficiently serious attention to the major point that was being made which was that they have predominantly studied short term change in

weight. I suggested that this should be acknowledged but they have ignored this. It is noteworthy at least that the results are different in the small group of people with longer follow up.

The point here is that the take home message that some people will see is that TEE isn't related to weight gain. This has important potential public health implications if true. However most people reading that conclusion will assume it is based on rather longer follow up than less than 4 weeks. The authors cannot conclude that people with higher TEE are more or less likely to gain weight over the longer term because they have so little data beyond 4 weeks. The data that they do have does tend to undermine their own conclusion.

Thus it seems to me to be important to make it clear in the title that TEE is not predictive of "short term" changes in body composition. It would also be necessary to make this much clearer in the conclusions.

Response: We discuss the suggestive finding in more detail in the conclusions:

"If greater TEE was protective against gaining fat, then subjects with greater adjusted TEE, or positive changes in adjusted TEE, should have experienced less weight and fat gain. Instead, we found a positive relationship between the difference in adjusted TEE between measurements and change in body weight (in both datasets), and no relationship between any measure of adjusted TEE (time 1, difference between measures, or average) and change in body fat percentage (in both datasets) among adults in our sample. The similarity between the results in both datasets indicates that these findings are not an artefact of measurement error and short time intervals. It is noteworthy that there was a trend towards a negative relationship between adjusted TEE1 and change in body weight (P= 0.094) in the subset of individuals for which the period between repeated TEE measurements was longer than 4 weeks, but none of the other relationships indicated that TEE is predictive of changes in body weight or composition in this subset. Therefore, we cannot rule out the possibility that TEE would be predictive of small changes in body weight over much longer timeframes. But the results of the current study are consistent with those of previous work conducted on adults^{13,14} and children⁹⁻¹² which reported no relationship between TEE and change in body fat percentage." (lines 406-423)

However, we disagree with the reviewer and do not think that the one suggestive result warrants adding "short-term" in the title because none of the other results indicated a negative relationship between adjusted TEE and change in body weight or body fat percentage. When using the subset, we found no relationship between adjusted TEE1 and body fat percentage, between average adjusted TEE and body weight or body fat percentage, and the difference in adjusted TEE between measurements also did not predict body fat percentage, and was positively associated with changes in body weight (supporting the results from the analyses including all data).

Reviewers' Comments:

Reviewer #2:

Remarks to the Author:

This is a revision of a manuscript that I have reviewed before. The comments have been well addressed overall. However, I do agree with Reviewer 4 that the analysis of a correlation between body weight change and FM is largely based on short-term data. Even the cutoff of 4 weeks is still quite short as compared to human lifetime. I think this limitation could be more clearly stated (maybe not necessarily in the title but in manuscript and abstract). I am also wondering if a cutoff of 1 year would be suitable, though there is probably not enough data in that range.

L381: Suggest to add $h^2 = 0.11$ for clarity.

L449: Stretching the interpretation to entire human (adult) lifespan seems quite daring given that the maximum for repeated measurements is 8 years and most of the repeated measurements are less than 4 years apart.

L490: Repeatabilities are tested at the boundary and tests are therefore typically one-tailed.

L548: Suggest: "We fitted the fixed effect..."

References 32 and 52 are identical.

Supplement: Suggest to spell adjusted TEE with lowercase a for consistency.

Reviewer #4:

Remarks to the Author:

I think that the authors' dismissal of the suggestion that the short-term nature of the follow-up of body composition in this study should be made explicit is unreasonable.

It is a fact that most of the data comes from studies of people with less than 4 weeks follow-up. It is not a matter of opinion.

The authors themselves state in their response that "there was a trend towards a negative relationship between adjusted TEE1 and change in body weight ($P = 0.094$) in the subset of individuals for which the period between repeated TEE measurements was longer than 4 weeks". This is also a fact.

They also state that "we cannot rule out the possibility that TEE would be predictive of small changes in body weight over much longer timeframes". This is also true.

Thus I don't see why the authors are so adamant that they won't include mention of the short-term nature of most of the change data in the title and the abstract. Their reasoning that "none of the other results indicated a negative relationship" is not really the issue. The fact is that this is mostly a short-term follow-up analysis and the title and abstract should reflect that truth.

REVIEWER COMMENTS

Reviewer #2 (Remarks to the Author):

This is a revision of a manuscript that I have reviewed before. The comments have been well addressed overall. However, I do agree with Reviewer 4 that the analysis of a correlation between body weight change and FM is largely based on short-term data. Even the cutoff of 4 weeks is still quite short as compared to human lifetime. I think this limitation could be more clearly stated (maybe not necessarily in the title but in manuscript and abstract). I am also wondering if a cutoff of 1 year would be suitable, though there is probably not enough data in that range.

Response: Yes, there would not be enough data if we were to use a cutoff of 1 year.

In response to both Reviewers we have explicitly specified the time intervals analyzed in the Abstract (lines 142 and 145) and we have highlighted more often throughout the text that most of the repeated measurements were taken within short periods of time (see also response to the comments by Reviewer 4).

L381: Suggest to add $h^2 = 0.11$ for clarity.

Response: We followed this suggestion and included $h^2 = 0.11$:

“A sibling study in humans (37 siblings aged 5 – 9 y) reported a low heritability ($h^2 = 0.11$) of TEE adjusted for resting metabolic rate³⁸.” (line 344-346)

L449: Stretching the interpretation to entire human (adult) lifespan seems quite daring given that the maximum for repeated measurements is 8 years and most of the repeated measurements are less than 4 years apart.

Response: We removed “and possibly throughout the adult lifespan” from the sentence:

“Our analyses here show that having a “fast” or “slow” metabolism is a repeatable, durable trait for adults that is consistent over years.” (lines 401-403)

L490: Repeatabilities are tested at the boundary and tests are therefore typically one-tailed.

Response: We included this text in the *Statistical analysis section*:

“Repeatabilities are tested at the boundary and tests are therefore typically one-tailed, and all other tests were two-tailed.” (lines 441-442)

L548: Suggest: “We fitted the fixed effect...”

Response: We changed the text following this suggestion.

References 32 and 52 are identical.

Response: Thank you for spotting this error. We remove the duplicated reference.

Supplement: Suggest to spell adjusted TEE with lowercase a for consistency.

Response: We followed this suggestion.

Reviewer #4 (Remarks to the Author):

I think that the authors dismissal of the suggestion that the short term nature of the follow-up of body composition in this study should be made explicit is unreasonable.

It is a fact that most of the data comes from studies of people with less than 4 weeks follow up. It is not a matter of opinion.

The authors themselves state in their response that "there was a trend towards a negative relationship between adjusted TEE1 and change in body weight (P= 0.094) in the subset of individuals for which the period between repeated TEE measurements was longer than 4 weeks". This is also a fact.

They also state that "we cannot rule out the possibility that TEE would be predictive of small changes in body weight over much longer timeframes". This is also true.

Thus I don't see why the authors are so adamant that they won't include mention of the short term nature of most of the change data in the title and the abstract. Their reasoning that "none of the other results indicated a negative relationship" is not really the issue. The fact is that this is mostly a short term follow up analysis and the title and abstract should reflect that truth.

We understand the Reviewer's perspective and agree that we should be clear about the timeframes analyzed. Rather than a relative or subjective term such as "short", in response to comments by both reviewers we have explicitly specified the time intervals analyzed in the Abstract and we have highlighted more often in the text that most of the repeated measurements were taken within short periods of time:

In lines 136, 142 and 145 in the abstract, and lines 226-228, 280, 356 and 465 in the manuscript.

We respectfully disagree that it would improve the title to use the term "short" or "short term" because 1) a substantive portion of the analyses include longer intervals and 2) "short" is ambiguous. Instead, specifying the intervals in the Abstract provides an objective measure that specifies the nature of the analyses for the reader.

However, we are happy to defer to the Editor's preference for the title.